# Laser-Formed Sensors with Electrically Conductive MWCNT Networks for Gesture Recognition Applications

**DOI:** 10.3390/mi14061106

**Published:** 2023-05-24

**Authors:** Natalia A. Nikitina, Dmitry I. Ryabkin, Victoria V. Suchkova, Artem V. Kuksin, Evgeny S. Pyankov, Levan P. Ichkitidze, Aleksey V. Maksimkin, Evgeny P. Kitsyuk, Ekaterina A. Gerasimenko, Dmitry V. Telyshev, Ivan Bobrinetskiy, Sergey V. Selishchev, Alexander Yu. Gerasimenko

**Affiliations:** 1Institute of Biomedical Systems, National Research University of Electronic Technology, 124498 Moscow, Russia; ryabkin@bms.zone (D.I.R.); molodykh1999@gmail.com (V.V.S.); nix007@mail.ru (A.V.K.); zugusik@gmail.com (E.S.P.); ichkitidze@bms.zone (L.P.I.); katball@mail.ru (E.A.G.); telyshev@bms.zone (D.V.T.); selishchev@bms.zone (S.V.S.); 2Institute for Bionic Technologies and Engineering, I.M. Sechenov First Moscow State Medical University, Bolshaya Pirogovskaya Street 2-4, 119991 Moscow, Russia; aleksey_maksimkin@mail.ru; 3Scientific-Manufacturing Complex “Technological Centre”, Shokin Square 1, bld. 7 off. 7237, 124498 Moscow, Russia; kitsyuk.e@gmail.com; 4Center for Probe Microscopy and Nanotechnology, National Research University of Electronic Technology, 124498 Moscow, Russia; vkn@nanotube.ru

**Keywords:** strain sensors, flexible electronics, smart wearable electronics, carbon nanotubes, electrical conductive networks, laser structuring, gesture recognition, sensor glove

## Abstract

Currently, an urgent need in the field of wearable electronics is the development of flexible sensors that can be attached to the human body to monitor various physiological indicators and movements. In this work, we propose a method for forming an electrically conductive network of multi-walled carbon nanotubes (MWCNT) in a matrix of silicone elastomer to make stretchable sensors sensitive to mechanical strain. The electrical conductivity and sensitivity characteristics of the sensor were improved by using laser exposure, through the effect of forming strong carbon nanotube (CNT) networks. The initial electrical resistance of the sensors obtained using laser technology was ~3 kOhm (in the absence of deformation) at a low concentration of nanotubes of 3 wt% in composition. For comparison, in a similar manufacturing process, but without laser exposure, the active material had significantly higher values of electrical resistance, which was ~19 kOhm in this case. The laser-fabricated sensors have a high tensile sensitivity (gauge factor ~10), linearity of >0.97, a low hysteresis of 2.4%, tensile strength of 963 kPa, and a fast strain response of 1 ms. The low Young’s modulus values of ~47 kPa and the high electrical and sensitivity characteristics of the sensors made it possible to fabricate a smart gesture recognition sensor system based on them, with a recognition accuracy of ~94%. Data reading and visualization were performed using the developed electronic unit based on the ATXMEGA8E5-AU microcontroller and software. The obtained results open great prospects for the application of flexible CNT sensors in intelligent wearable devices (IWDs) for medical and industrial applications.

## 1. Introduction

Hand gesture recognition has attracted widespread research interest as an important approach to human–machine interaction. Enabling direct human–computer communication is necessary in many areas: for people with disabilities [1], in virtual reality [2], video games [3], and robotics [4]. Studies have found that regular training of the hands in the form of monotonous, purposeful repetition of specific movements (gestures), leads to a reorganization of the cerebral cortex and the formation of a permanent therapeutic effect on the patients’ motor activity recovery [5]. The effectiveness of training depends directly on the accuracy of the gesture recognition. In selecting the best method of gesture detection, it is important to consider that the human hand has a complex, asymmetrical geometric shape and that hand gestures are unpredictable and varied. In this regard, the following limitations should be addressed in constructing a gesture recognition system: segmentation of the gesturing hand, and a selection of specific purposeful gestures in a continuous stream of chaotic body movements.

One of the most popular approaches to recognizing hand gestures is visual. Visual gesture recognition methods use a combination of digital cameras, visible and infrared range to capture finger movement and further interpretation, so gesture reading is performed without the use of additional connected sensors [6]. The visual approach requires the use of a complex, extensive mathematical apparatus to construct a three-dimensional model of the hand [7]. Besides the listed disadvantages, the visual method of gesture recognition is the most expensive because of the high-tech assistive equipment used.

Radar gesture recognition technology is less sensitive to external noise [8,9]. The range resolution for hand gesture recognition is determined by the bandwidth of the radio waves. This approach requires the development of expensive equipment to generate radio waves with a sufficiently large bandwidth to provide the necessary resolution.

Ultrasound is well known for the ability to penetrate several cm under the skin and deliver information about both superficial and deep muscle tissues. This feature allows for the use of ultrasonic sensors for gesture recognition [10]. Recognition is based on an ultrasound image constructed from the propagation parameters of ultrasound waves [11]. This approach allows for a more miniaturized gesture recognition system. However, the accuracy of the ultrasonic gesture recognition system remains insufficient.

One of the most cost-effective approaches to gesture recognition is the use of accelerometers and gyroscopic sensors [12]. Most of these systems use a gyroscope, which reads angular velocity as the hand gestures change and transmits it to a personal computer interfaced to the three-axis gyroscope. Sensor data is processed using a moving average filter to reduce noise [13]. Gyroscopic sensors can only indirectly estimate the position of the finger joints, so the accuracy of movement registration is not high enough.

A promising approach to recording hand movements and gestures is the use of flexible strain sensors. Traditional microelectronic strain sensors, by the rigidity of their construction, cannot provide the necessary response on flat and/or curved surfaces with large deformations, such as the human body [14]. In contrast, the development of flexible strain sensors makes it possible to measure hand gesture movements directly by placing the sensors on the fingers or integrating them into gloves for data transfer [15]. The basic principle of flexible strain sensors is the generation of electrical signals (resistance, capacitance, voltage, etc., depending on the type of active material) in response to applied strain [16]. Capacitive strain sensors demonstrate good linearity, low hysteresis, and high sensitivity [17]. However, the variation in capacitance is typically in the order of pF and requires quite sensitive measuring equipment. In addition, capacitive sensors have some disadvantages, such as high impedance, making them sensitive to the stray capacitance of coaxial cables and leading to corresponding measurement errors [18]. Other types of sensors, such as Bragg fiber array [19], triboelectric [20], and piezoelectric [21] sensors, usually cannot detect slow or static strain due to rapid charge transfer. In general, they cannot detect slow or static deformation due to rapid charge transfer. In addition, their practical application as skin-worn devices remains difficult due to the need for sophisticated measurement equipment. Resistive sensors are the most used [22,23] due to the lack of the need for complex measuring equipment and the ease of manufacturing. Traditionally, due to their simple design, high sensitivity, and linearity, piezoresistive sensors have been the most widely used in commercial microelectromechanical systems [24]. Current focus is placed on the development and improvement of flexible resistive sensors, as well as extending their range of applications.

The active materials of resistive sensors are mostly conductive micro/nano-polymer composites, thin films, or conductive filaments/fabrics [25]. Recent research and development of resistance sensors based on nanostructures and their composites have accelerated following the discovery of materials such as carbon nanotubes (CNT) [26]. CNT based nanocomposites and films are a promising alternative to other smart materials, largely owing to their superior electrical and mechanical properties, ultra-high intrinsic piezoresistivity caused by nanotube chirality [27], unique morphology, ease of manipulation, and chemical versatility. A significant correlation was found between the reversible mechanical strain and the electrical resistivity of single- and multi-walled carbon nanotubes (SWCNT and MWCNT). The correlation demonstrates that CNT can be used as precise nanoscale strain sensors [26]. Moreover, the addition of low concentrations of CNT to polymers provides the materials with piezoresistive properties. The obtained materials show a sufficiently linear and reversible change in electrical resistance under deformation [15,28]. The strain sensitivity of such CNT composites is based on the modification of the percolation network of nanotubes under mechanical action [29]. Recent results have shown that the characteristics of such strain sensors are closely related to the assembly methods and microstructure of the conductive networks [30]. Therefore, to manufacture high-performance flexible strain sensors based on CNT composites, it is crucial to be able to control the formation of the conductive network of nanotubes in the material during production.

Laser exposure is a universal and inexpensive method for modifying and assembling polymer composites [31,32,33,34,35]. Laser exposure of composites with CNT can significantly increase electrical conductivity by increasing the concentration of the carbon filler until the electrical conductivity percolation threshold is reached [36]. Laser exposure causes pyrolysis of the polymer matrix, which leads to the formation of gaseous particles leaving the material, thereby increasing the conductive filler/matrix ratio [37]. This approach makes it possible to make inexpensive materials, since it allows for the use of sufficiently low concentrations of expensive fillers, such as CNT. The morphology and structure of three-dimensional CNT networks are known to be controllable by adjusting the power and laser exposure time [38]. Thus, laser exposure of electrically conductive composites allows for the formation of conductive CNT networks with a specified morphology and improved electrical conductivity. In the process of the exposure of composites, laser energy is absorbed by electrons and transferred to CNT atoms. As a result, branched CNT networks are formed. It has been previously demonstrated that laser exposure can be used to form strong conductive networks of nanotubes and a carbon framework in polymer composites under normal atmospheres, with minimal costs and minimal losses [39,40,41,42]. At the same time, laser exposure allows for increasing the electrical conductivity and mechanical hardness of the final material. Laser exposure through improved binding of the nanotubes to each other has resulted in a low electrical resistance value.

In summary, this paper describes the manufacturing of flexible strain sensors consisting of electrically conductive networks of carbon nanotubes in polydimethylsiloxane (PDMS) elastomer. Laser processing has been applied to fabricate various flexible sensors [43,44,45]. The novelty of the project lies in the use of laser exposure to form electrically conductive networks of carbon nanotubes for the manufacture of gesture-registering strain sensors. This approach significantly increases the conductivity and sensitivity of the strain sensors, allowing for the use of lower MWCNT concentrations. Thus, the formed percolation network of MWCNT is more electrically sensitive to mechanical deformations. Under laser exposure, electrically conductive networks of carbon nanotubes were formed in the PDMS matrix. This has resulted in a high sensitivity of the sensors while maintaining high elongation and strength. Based on the sensors, a touch-sensitive smart system for hand gesture recognition in the form of a glove can be fabricated. Such a smart system is one of the reliable and inexpensive approaches, endowing a person with objective feedback during training of hand motor functions in rehabilitation after severe diseases and during learning.

## 2. Materials and Methods

### 2.1. Components Used for Flexible Sensor Manufacturing

The sensors are nanocomposites based on multi-walled carbon nanotubes (MWCNT), where the MWCNT have been encapsulated in a PDMS matrix. MWCNT (NanoTechCenter LLC, Tambov, Russia) in powder form were obtained via CVD synthesis, had an outer diameter of 8–30 nm, inner diameter of 5–15 nm, length ≥20 μm, specific surface area ≥270 m^2^/g, and bulk density 0.025–0.06 g/cm^3^. The two-component PDMS elastomer Ecoflex 00–10 (Smooth-On Inc., Macungie, PA, USA) had a dynamic viscosity in the mixed state of 14 Pa∙s, Young’s modulus at 100% elongation 0.06 MPa, hardness (Shore A) 10, density 1.04 g/cm^3^, with an operating range of 19 °C to 232 °C. Carbon fiber wires were used to connect the sensor.

The basis of the electronic unit to read and process signals from the sensors was a 10-bit microcontroller ATXMEGA8E5-AU (Microchip Technology Inc., Chandler, AZ, USA), enclosed in ABS plastic.

### 2.2. Flexible Sensor Fabrication

The sensors were produced by screen-printing with laser-scanning of the surface to form a strong electrically conductive network of MWCNT. The fabrication process is shown schematically in Figure 1. To compare the effect of laser radiation on the electrical properties, sensors without laser exposure were also fabricated. Sensors that were not laser-exposed were also produced. The manufacturing process was similar.

First, components A and B of the PDMS elastomer in the liquid phase were added to the carbon nanotubes in equal proportions and thoroughly mixed mechanically for 10 min. The concentration of MWCNT in the resulting mixture was 3 wt%. This percentage of nanotubes corresponded to the experimentally obtained percolation threshold, above which no significant changes in the electrical conductivity of the MWCNT/PDMS composite were observed. Then, the mixture was transferred to an XFL020 vacuum furnace (France Etuves, Chelles, France) and the degassing process was started (vacuum). After removing the air formed in the mixing process, the mixture MWCNT/PDMS was applied to the substrate via a U-shape screen. The forms printed using a DLP three-dimensional printer were used as a substrate and screen. The sensor dimensions can be adjusted by selecting a new shape of the screen with the required dimensions. The U-shape form of the sensors was chosen for the convenience of the wires which were added to the MWCNT/PDMS composite before the PDMS solidified. In addition, the U-shaped design has a longer active material length than the I-shaped sensor. This provides a greater change in resistance when deformed. The wires were placed on one end of the sensor due to the chosen shape. The thickness of the formed active material layer was 0.5 mm. This thickness is mainly due to the magnitude of the laser radiation drag to form an electrically conductive nanostructured layer. To increase the tensile strength and insulation of the conductive layer, the sensor was coated with 0.25 mm thick layers of PDMS on the top and bottom. The total thickness of the sensor was therefore 1 mm. After the active layer solidified, it was exposed to laser radiation at selected parameters. Pulsed Yb laser was used and a laser exposure pattern was set in the software (Figure 2a). The parameters of the laser exposure are shown in Table 1. Figure 2b shows the pattern under the pilot laser beam. Laser scanning was performed over the entire surface of the MWCNT/PDMS composite in a U-shape. After the MWCNT/PDMS layer was screen-printed and laser-treated, a thin layer of PDMS was poured on top through another rectangular screen. The active material layer with the PDMS-insulating layer was then inverted and the PDMS layer was poured on the other side in the same mold.

Laser exposure has been used to improve electrical performance and sensitivity due to the previously demonstrated effect of forming strong MWCNT networks with an increased electrical conductivity of the material [39,40,46]. This includes the proven effect of the enhanced electrical conductivity in various polymer and biopolymer matrices [31,32,33,34,41,42]. This approach allows for the use of low MWCNT concentrations (3 wt%) without the loss of sensitivity at large strains.

After laser-processing, the active sensor material (a U-shaped MWCNT/PDMS composite) was coated on both sides with a layer of pure PDMS to insulate it from external damage. The sensors made by this technique consisted of three layers, as demonstrated in Figure 3. The inner layer of the MWCNT/PDMS is the active material of the sensor, ensuring strain sensitivity. The MWCNT/PDMS layer was coated with layers of pure PDMS on both sides.

### 2.3. Sensitivity Study of the Sensors

The study of the sensitivity of the developed sensors included the construction of curves of dependence of electrical resistance on the applied strain and the calculation of the gauge factor (GF).

To plot the curves of electrical resistance dependence on the applied strain, the sensors were connected to a multimeter (UNI-T UT33C+, Rexant International Ltd., Hong Kong, China) and the resistance values were recorded from 0 to 100% stretch (readings were taken every 0.5 cm of elongation).

GF is defined as the ratio of the relative change in electrical resistance to the applied strain. The range of strain for GF determination was 0–100%. This range is related to the fact that the finger bending movement can result in a strain n of less than 100% [47]. The GF of the developed sensors was calculated using the equation [30]:GF = (ΔR/R_0_)/(Δl/l_0_),(1)
where ΔR and Δl are the absolute change in electrical resistance and length, respectively, under strain n, R_0_ and l_0_, which are the initial value of electrical resistance and length, respectively.

Resistance hysteresis, a typical shortcoming of most resistive sensors, was estimated using the equation [48]:h (%) = ((Rs − Rr)/(R_max_ − R_0_)) × 100%,(2)
where R_s_ and R_r_ are the resistance during stretching and releasing, respectively, at the same strains, R_0_, which is the initial (minimum) value of resistance, R_max_, the maximum resistance value.

### 2.4. Linearity Study of the Sensors

Linearity is the coefficient of determination of GF under stretching up to 100% of the length. Sensor linearity was calculated using the equation:R^2^ = *∑*(GF_predicted_ − GF_mean_)^2^/*∑*(GF_actual_ − GF_mean_)^2^,(3)
where R^2^ is the determination coefficient, GF_predicted_ is the predicted GF value, GF_mean_ is the average GF value, and GF_actual_ is the actual calculated GF value. The predicted GF values were determined by constructing a linear regression using the sklearn library in the python programming language.

### 2.5. Study of Mechanical Characteristics of the Sensors

The sensor elasticity should correspond to the elasticity of human skin in order to accurately replicate the movements during measurements. The elasticity (Young’s modulus) was calculated using the equation:E = F·l_0_/S·∆l,(4)
where F is the applied force, l_0_ is the initial length, S is the area, and ∆l is the absolute change in length because of the applied strain.

### 2.6. Cyclic Sensor Testing

To confirm the reliability of the developed sensors, cyclic tests were conducted for one month. The sensors were subjected to daily cyclic load: percentage elongation ε = 100% and relaxation to the initial length. The number of repetitions per cycle was 10,000. During the test, the resistance values were monitored at the starting point ε = 0% and the end point ε = 100%. The measurements were carried out with a testing machine equipped with a positioning system. For testing, the strain sensor was gripped by the clamps integrated in the moving head of the test machine. To control the resistance, the strain sensor wires were connected to a multimeter. Using a specially developed software, the positioning system has been programmed to stretch the strain sensor for 1 s to ε = 100%, then return to the initial length of the sensor.

### 2.7. Manufacturing of an Electronic Signal-Reading Unit

The developed resistive strain sensors convert external deformation into an electrical signal. Their structure contains an electrically conductive active material connected to a flexible substrate. A change in the structure of the active material under applied strain results in a change in electrical resistance. In fact, the conductive network of active materials of the developed sensor acts as a variable resistor under the applied voltage. On this basis, a specialized electronic unit was developed to read the signals of the developed flexible sensors.

The electronic unit was developed as follows: based on the resistance values of the developed sensors, an electrical circuit was simulated using the LTspice software (Analog Devices Inc., Wilmington, NC, USA), and the values of the resistors used in the circuit were selected. The developed electrical circuit is shown in Figure 4a. Next, the circuit for converting the electrical resistance of the sensors into voltage was calculated using a 10-bit ATXMEGA8E5-AU microcontroller (Atmel Corporation, San Jose, CA, USA). The input voltage range applied to the ADC was 0 to 1.14 V. The analog circuit was designed so that the voltage converted by the LM358D operational amplifier (STMicroelectronics, Geneva, Switzerland) from the sensor resistance is in the input voltage range. In this case the sensor acts as a variable resistor. In the electrical circuit, R4 is designated (Figure 4a). In accordance with the developed circuit, an electronic unit was constructed from electronic components (Figure 4b). The electronic circuit (1) was based on the ATXMEGA8E5-AU microcontroller, a Bluetooth module (2) was added to transfer data to a computer with the software installed, a power button (3) allowed for starting the unit, wires (4) provided a sensor connection, and a USB connector (5) was needed to connect to a 5 V power supply or to a computer. Finally, the case was closed with a lid.

### 2.8. Manufacturing of the Smart System for Gesture Recognition

The flexible sensors made according to the method described above (in Section 2.1) were attached using conventional medical tapes to a soft, thin tissue glove. The sensors were attached to the glove at the middle mobile joints of the fingers and connected to the developed electronic unit. No more than two resistive sensors could be connected to one electronic block. The electronic block was connected to a computer. The software for the electronic unit displayed the ADC values of each connected sensor on the computer screen and plotted the ADC values over time. The ADC values changed with finger movements, and the graph clearly represented the amplitude of the movements. Numerical ADC values were recorded and stored for further processing and machine learning of hand gesture recognition. The data were recorded at intervals of 60 values per second. A schematic representation of the developed smart system for gesture recognition is shown in Figure 5.

### 2.9. Gesture Recognition Techniques Using a Smart System and Machine Learning

Gesture recognition is a multiclass classification task. The One-vs-Rest approach was used to solve the problem. The One-vs-Rest approach trains n binary classifiers, where n is the number of classes. Each classifier predicts whether an instance belongs to one of n classes or not. The class of which the probability of belonging was the most probable was chosen.

The K nearest neighbor classifier (KNN) and support vector machine (SVM) models, as well as ensembles over decision trees random forest classifier (RF), xgboost classifier (XGB), LightGBM Classifier (LGBM), decision tree classifier (DT), logistic regression (LR) were used for gesture recognition. The ML models KNN, SVM, RF, DT, LR from sklearn (ver. 0.23.2), XGB from xgboost (ver. 1.4.1), and LGBM from LightGB (ver. 3.3.2) were used.

Accuracy was chosen as the quality metric for the gesture recognition task. The accuracy metric for multiclass classification is the ratio of the number of correct predictions to the total number of predictions. With a significant class imbalance, the accuracy metric may show incorrect results. When generating the dataset of resistive sensors, the exposure time of the gestures was equal, thereby achieving a balance in the information for the different gestures.

The formation of the dataset was carried out automatically. Resistive sensors were connected to the electronic unit, which was connected to a personal computer with special software. The software recorded the date, time, resistance values received from the sensors, as well as the demonstrated gesture into a .csv file. The person demonstrated the gesture according to the information from the software. The demonstrated gesture was changed every 5 s. This way, automated data labeling was performed. A maximum of two resistive sensors could be connected to one electronic unit. In this way, the data are written to 3 files. When combining data from files by a sensor from different fingers, the different sampling rate of data recording and the start and end time of the recording are considered. 

The optimization of the hyperparameters of models with the best values of quality metrics was performed using the Optuna package (ver. 2.10.0). The data were modified with a moving average filter for better training quality. After filtering, the data were normalized using MinMax Scaler.

## 3. Results and Discussion

### 3.1. Structure and Initial Characteristics of the Sensors

Flexible electronic devices are gaining enormous popularity for wearable applications. The flexible sensors manufactured in this paper are the interface component of a smart system being developed to record hand gestures. Essentially, the developed sensors are a flexible resistor that changes its resistance under strain, so the sensors can be useful in many wearable electronics applications.

The sensors manufactured in this work (Figure 6) were a three-layer nanocomposite with dimensions of 35 × 15 × 1 mm. The active layer of sensors, i.e., the MWCNT/PDMS composite had dimensions of 30 × 12 × 0.5 mm (Figure 6d). Flexibility and stretchability of the sensors was provided by the constituent polymer, PDMS elastomer. The deformation sensitivity was provided by the included carbon nanotubes.

In order to compare the electrical and mechanical properties, the sensors were manufactured in two ways: with and without the use of laser exposure.

The structure of the network of conductive particles in polymer composites is known to determine their electrical properties [49]. Considering this, a laser exposure was applied in the sensor fabrication process to form an electrically conductive network of MWCNT by forming bonds between the individual nanotubes. Firstly, laser exposure of carbon nanotube films has previously been found to cause carbon nanotubes to bind and weld together, resulting in strong electrically conductive networks [34,35]. In polymer matrices, laser nanotube network formation not only increases electrical conductivity, but also increases material strength [50].

Figure 7 shows the structure of the electrically conductive network of MWCNT sensors obtained without laser exposure (simple air-drying) and with laser exposure. Images were obtained through scanning electron microscopy (SEM), using an FEI Helios NanoLab 650 microscope (FEI Company, Hillsborough, OR, USA) with an electron column acceleration voltage of 1 kV and an electron probe current of 21 pA. The internal structure was studied for sensors made without laser exposure (Figure 7a) and with laser exposure (Figure 7b). After laser exposure, a redistribution of the MWCNT clusters is observed. Large clusters are separated into smaller clusters, increasing the number of clusters and decreasing their average size. The electrically conductive network of MWCNT after laser exposure has become less complex, with predominantly nanotube bonding areas. A sparser network is preferable for strain sensors because it allows for a lower resistance hysteresis. This is because a sparser network stretches more uniformly and the rearrangement of the conductive network becomes more stable due to the absence of a large number of conductive paths. This means that there is less chance of random contribution to the conductivity of untangled or randomly connected nanotubes. Figure 7c,d shows the areas where the nanotubes are welded together under higher magnification. The arrows show the contact points between the MWCNT.

The electrical properties of the sensors obtained with and without laser exposure were compared using different concentrations of 2, 3, 4 wt% of nanotubes (Figure 8). Increasing the MWCNT concentration above 4 wt% significantly increases the viscosity of the MWCNT/PDMS blend. This makes it difficult to homogenize the MWCNT in PDMS during the fabrication of the active material. Concentrations of CNT below 2 wt% will cause the sensor resistance to rise above 1 MOhm. Such high resistance values result in difficulty in detecting changes in resistance and are unacceptable for sensor construction. If 2% is exposed to higher laser power, the silicone in the active sensor layer will overheat and burn out. This will reduce the performance of the sensor.

The initial electrical resistance of the laser-produced sensors was ~3 kOhm (no deformation, Figure 8) at a low concentration of 3 wt% nanotubes in the composition. In comparison, using a similar fabrication process without laser exposure, the active material had significantly higher electrical resistivity values, in this case ~19 kOhm. Increasing the concentration of the nanotubes resulted in a slight increase in strength, which is related to the high strength of the nanotubes themselves, allowing them to be used as a reinforcing material.

### 3.2. Strain Response

High strain sensitivity of traditional semiconductor and metal sensors is achieved with a small range of relative strain <3% and only in a certain direction [51]. These types of devices are widely used as “rigid” sensors because they have low flexibility and extensibility, so they are designed to be attached to rigid materials (concrete, metal, plastic) to monitor the condition of the structure or quantify the small strain of the sample. The current demand for wearable electronic devices has changed the structure of resistive type strain sensors from fragile to tensile. For resistive sensors, the working mechanism is the change in electrical resistance in response to tensile (bending) and compressive strains. A common indicator is sensitivity, which is quantified through the GF.

The strain response of the developed sensors obtained with the two methods (for comparison) was evaluated under tensile stress from 0 to 100%. The developed sensors respond to the tensile strain as follows. The electrical resistance of the sensors increases during stretching and decreases during return to the initial state. By recording the changes in electrical resistance, it is possible to judge the magnitude of elongation. The strain response time was 1 ms. Figure 9 shows the plots of the relative change in electrical resistance when the sensors are extended (Figure 9a) and released (Figure 9b). The relative change in resistance is presented as a percentage. The resistance hysteresis is shown in Figure 9c. The sensitivity coefficient of the sensors at different strains is shown in Figure 9d. It should be noted that the sensors exposed to laser exposure showed a higher GF (Figure 9c).

It was found that the sensors obtained using the laser-formed conductive network method exhibit a wider variation in electrical resistance under deformation (Figure 9a,b blue lines). This is apparently related to the efficient formation of an electrically conductive network of nanotubes in the sensor material, which conducts an electric current more efficiently. For further research and fabrication of the smart system, laser produced sensors were used, as they showed a higher sensitivity coefficient (Figure 9c). Their average sensitivity coefficient is 10, with a maximum value of 15.4 at 100% strain. Sensitivity is strain dependent and increases with increasing strain range.

The presence of hysteresis is a typical problem of flexible sensors. Hysteresis is related to the viscoelastic nature of flexible polymers and the slip of internal conductive nanomaterials [52,53]. As a result, the conductive network needs time to recover. The calculated hysteresis value of the sensors obtained with laser exposure was 2.4%, while that without laser exposure was 5%. The lower hysteresis of the laser-exposed sensors (Figure 9d) indicates a faster recovery of the conductive network. The lower hysteresis values can be explained by the efficient formation of the conductive network.

Although laser exposure increases the strength of materials, the laser-assisted sensors are flexible enough to follow movements accurately and without constraint. The Young’s modulus of the laser-irradiated sensors was 47 kPa, which is comparable to the Young’s modulus of human skin, which is 25–250 kPa [54]. Table 2 shows a comparison of the sensor characteristics investigated in this paper.

### 3.3. Linearity and Cyclicality

Laser-formed sensors were selected for the fabrication of an intelligent gesture recognition system. The strain response speed and linearity of these sensors were evaluated. The processing speed of the output signal directly depends on the linearity of the sensor GF. The key feature of using flexible polymer/nanoparticle-based strain sensors for gesture recognition is the presence of two strain linearity areas ε = 0–10% and ε > 10%, caused by hysteresis [25]. Figure 10 compares the actual GF values with the predicted GF using linear regression. The linearity for the 0 to 10% stretch region (Figure 10a) reached 0.998, and for the 10% to 100% stretch region (Figure 10b) 0.97. Since in finger bending, the stretch of the sensors usually does not exceed 10%, the area with the highest linearity is mainly used, which determines the high accuracy and predictability of the results.

Developed sensors, due to their flexibility, present the possibility to measure deformations of body parts caused by anatomical movements. By attaching the sensor to the area under investigation, it is possible to determine strain directly in the area under investigation. The resistance of the developed sensors changes due to the restructuring of the formed network of carbon nanotubes. By recording the resistance changes of the developed sensors, the registration of strain is realized. Resistance changes during strain are due to (1) geometric changes (length and cross-sectional area of the active material) [52], (2) separation of the tunnel gaps due to strain and reconstructing of the conductive network [29], (3) the intrinsic piezoresistive properties of carbon nanotubes [27]. When the deformation is removed, the resistance is restored as the conductive network of nanotubes returns to its original morphology.

The results of the study demonstrated that regular loads do not affect the electrical characteristics of the sensors negatively and, accordingly, the sensors can ensure a stable operation of the smart system. The resistance fluctuations remained within the hysteresis range (Figure 11). For 30 days and a total of 300,000 cycles, no degradation of the conductivity and decreased sensitivity of the sensors was observed. This repeatability is caused using the proposed laser processing of the active material of the sensors. The high strength of the created networks of carbon nanotubes makes it possible to keep the conductive properties of the active material for a long period of time.

### 3.4. Gesture Recognition

Sensors were manufactured for each finger of the hand (Figure 12a). Sensors were attached to each finger in the moving joints using medical tapes to a soft fabric glove. The glove was comfortable to wear, fit the fingers tightly, and did not restrict movement. The sensors were connected to the developed electronic unit, the appearance of which is shown in Figure 12b. Then, the electronic unit was connected to a personal computer by means of a wireless Bluetooth connection. The appearance of the resulting smart system for gesture recognition is shown in Figure 12c. The program for visualization, reading and storing data from the sensors developed for the electronic unit was run on the computer. The visualization software plotted the amplitude of the ADC values over time. A screenshot of the program window content display by the PC is shown in Figure 12d. The sensors contain wires for connection to the electronic unit. The function of the wires is to transmit the signal from the active sensor layer to the electronic unit. The electronic unit can be powered by a USB cable or a battery charger.

To train the ML model for gesture recognition, a dataset was formed. The subject demonstrated in turn the basic and recognizable gesture. The subject wore a glove with sensors. The wires that come from the sensors were connected to the electronic unit to record the signals. The electronic unit was connected to a computer and the developed software run, which recorded the data received from the sensors over time in a separate file. Eleven different gestures were chosen for the gesture recognition experiment, including a basic gesture. The signals produced by the alternation between the base gesture and the studied gesture were recorded. The basic gesture was a hand clenched in a fist. For each gesture, data were collected for 30 min, the basic and recognized gestures changed every 5 s.

Figure 13 shows the plots of dependencies of resistive sensor values during the demonstration of various gestures.

The resistance value of the sensors is time-varying in long-term measurements. To improve the quality of the ML model predictions it is necessary to apply filtering to the sensor data. Filtering by the moving average was applied:(5)$$\hat{f}=\frac{\sum_{i=j-n}^{j-1}f_{i}+\sum_{i=j+1}^{j+n}f_{i}}{2n},$$
where f is the value of the feature received from the sensor, $\hat{f}$ is the value after filtering, j is the feature value number in dataset, n is half the width of the interval of values across which the average is calculated. Figure 14 shows graphs of the data as a function of time during the “three” gesture demonstration, before and after filtering. Figure 14 shows the application of filtering to the sensor signal. With prolonged operation, a change in the basic resistance of the sensor occurs. Then there is practically no change in baseline resistance. Filtering is used to correct for changes in the sensor resistance baseline (blue curves before filtering, orange after). Signal fluctuations in the orange curves are the result of changes in sensor resistance and are not noise.

The dataset was collected based on data from sensors detecting the amount of resistance when the subject displayed gestures. Eleven gestures were used in the data collection: 1 basic gesture (the fist), and 10 recognizable gestures. The screen showed information for the subject which gesture to demonstrate, while at the same time, the data from the sensors were written into a .csv with a label value corresponding to the information being shown. This is how the data were labeled. Each gesture was measured for ~30 min, with the displayed gesture changing every 5 s. Data were recorded with a discretization in the range of 20–60 Hz. The final dataset for model training contained over 560,000 lines.

Seven machine learning models were used for gesture recognition: K nearest neighbor classifier (KNN), support vector machine (SVM) random forest classifier (RF), xgboost classifier (XGB), LightGBM Classifier (LGBM), decision tree classifier (DT), logistic regression (LR). The accuracy metric was used as a classification quality metric. Since training several models on the entire dataset is an extremely resource-intensive task, to compare the quality, we used 10^5^ rows from the dataset. The following accuracy was obtained for the classifiers: 0.941 ± 0.001 (RF), 0.887 ± 0.001 (KNN), 0.604 ± 0.004 (SVM), 0.734 ± 0.005 (XGB), 0.731 ± 0.003 (LGBM), 0.753 ± 0.003 (DTC), 0.490 ± 0.005 (LR) (Figure 15a). Increasing the dataset size improves the prediction accuracy (Figure 15b). However, if the dataset is larger than 4.5 × 10^5^, the accuracy does not increase significantly.

For the best performing ML models, the RF and KNN hyperparameters were optimized. After hyperparameter optimization, the KNN model showed an accuracy of 0.965 ± 0.001, and the RF model showed an accuracy of 0.958 ± 0.002.

For wearable applications where significant dynamic loads are often encountered, durability is an important quality indicator. With long load cycles during data collection for machine learning, the flexible sensors built into the smart system ensured the reliable functionality of the developed system.

## 4. Conclusions

The effect of the formation of electrically conductive networks in the MWCNT/PDMS material under laser exposure is demonstrated. It is found that MWCNT/PDMS-based strain gauges obtained by laser structuring of conductive networks have lower initial resistance (3 kOhm), higher sensitivity (average GF 10) to strain and lower hysteresis (3%) of electrical resistance compared to those obtained without laser exposure. The Young’s modulus for the laser-irradiated sensors was 47 kPa, which is comparable to the Young’s modulus of human skin.

A sensor-based intelligent hand gesture recognition system based on laser-formed strain sensors has been successfully developed. The numerical values of the change in electrical resistance during finger movements were read and processed using the developed microcontroller electronic unit based on the ATXMEGA8E5-AU microcontroller. The data were transmitted wirelessly to a personal computer and displayed as a graph of the amplitude of movements versus time. The developed software allowed for recording the data in a convenient form for subsequent processing and machine learning for the purpose of machine gesture recognition. The use of a fabricated smart system based on laser-formed sensors with conductive MWCNT networks has achieved a gesture recognition accuracy of ~94%. The developed sensory smart system is one of the reliable and inexpensive approaches to provide objective feedback for training the motor functions of the hand.

## Figures and Tables

**Figure 1 micromachines-14-01106-f001:**
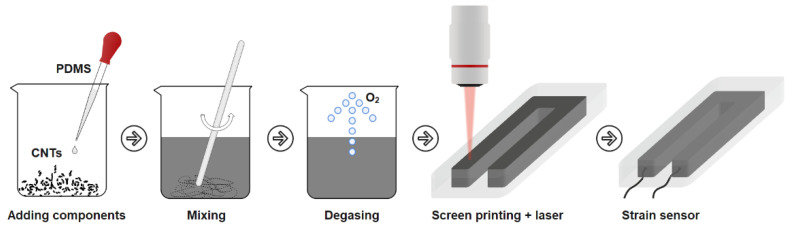
Schematic diagram of sensor manufacturing.

**Figure 2 micromachines-14-01106-f002:**
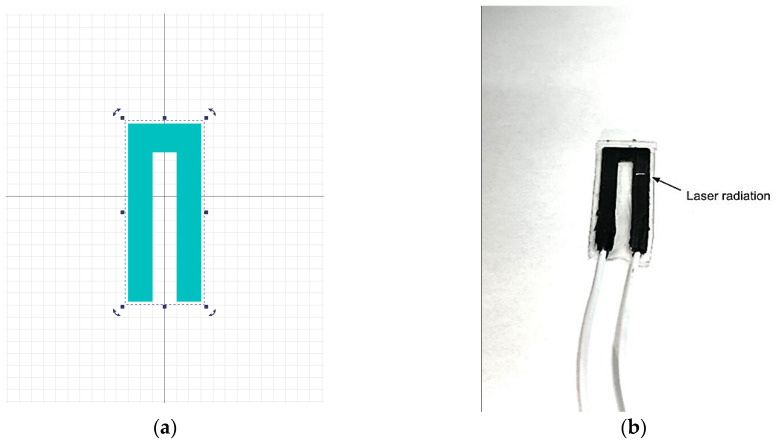
Laser exposure of the active material layer of the sensor: (**a**) template for laser scanning; (**b**) active material layer of the sensor in the process of laser exposure.

**Figure 3 micromachines-14-01106-f003:**
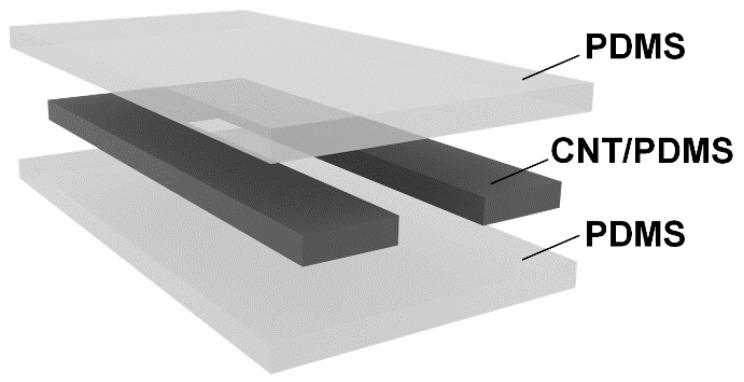
Schematic diagram of the manufactured sensor.

**Figure 4 micromachines-14-01106-f004:**
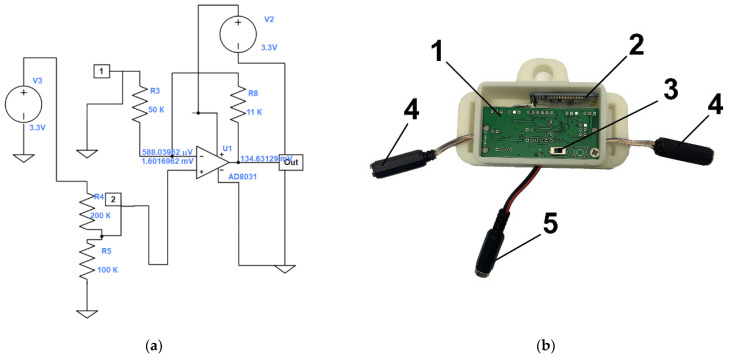
(**a**) Developed circuit design of the electronic sensor unit. (**b**) Electronic microcontroller circuit layout, where 1—electronic circuit for reading sensor signals, 2—Bluetooth wireless transmission module, 3—power button, 4—wires for sensor connection, and 5—power connector.

**Figure 5 micromachines-14-01106-f005:**
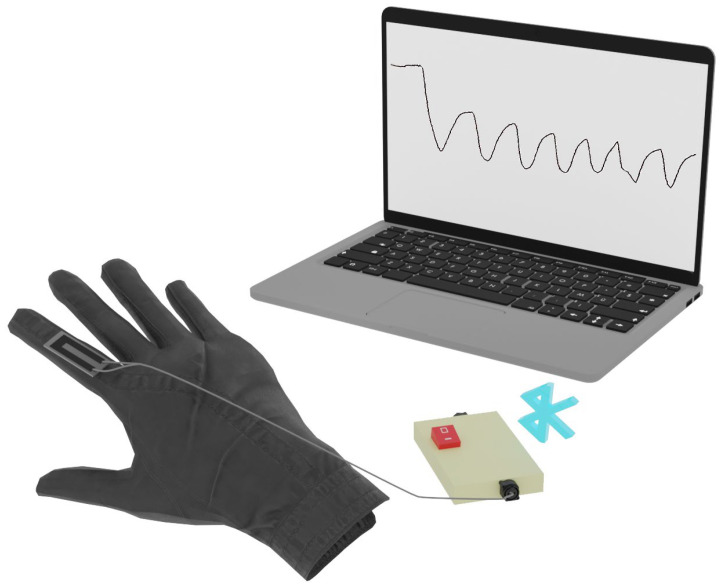
Schematic representation of a smart system for gesture recognition.

**Figure 6 micromachines-14-01106-f006:**
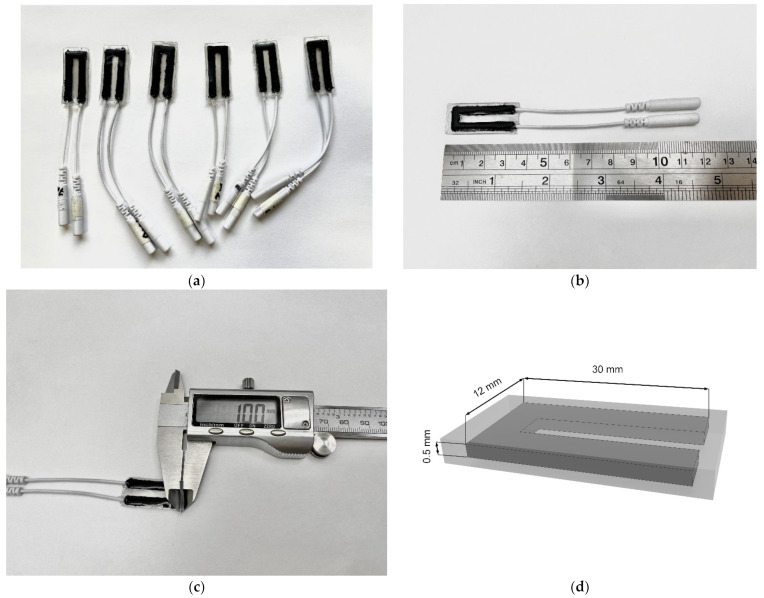
(**a**) Developed sensors based on carbon nanotubes and elastomers. (**b**,**c**) Photo of the sensor with the scale grid. (**d**) Dimensions of the manufactured sensor.

**Figure 7 micromachines-14-01106-f007:**
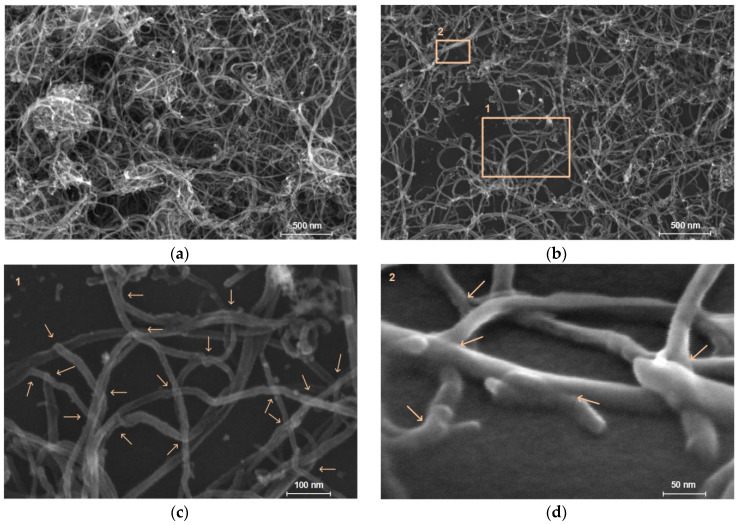
Internal structure of the sensors with enlargement mode ×120,000: (**a**) fabricated without laser exposure and (**b**) with laser exposure. Enlarged areas of laser-irradiated sensors: (**c**) Area 1 with enlargement mode ×500,000, (**d**) Area 2 with enlargement mode ×1,000,000 and at 52° angle. The arrows indicate the welded areas of the nanotubes formed by the laser exposure.

**Figure 8 micromachines-14-01106-f008:**
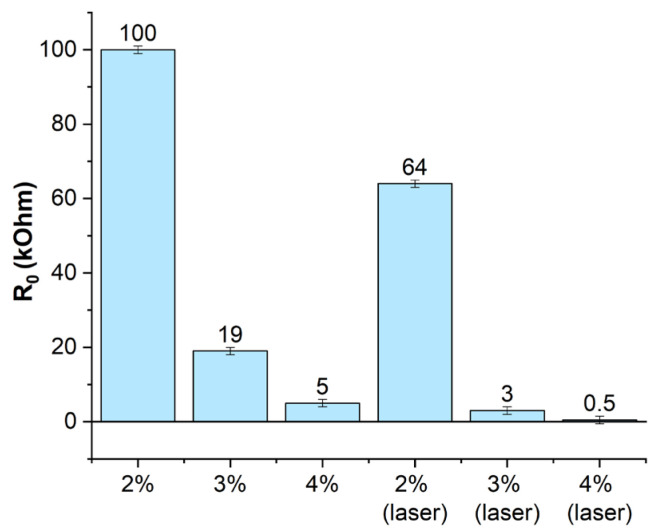
Comparison of the resistance of the active sensor material at different concentrations and fabrication methods.

**Figure 9 micromachines-14-01106-f009:**
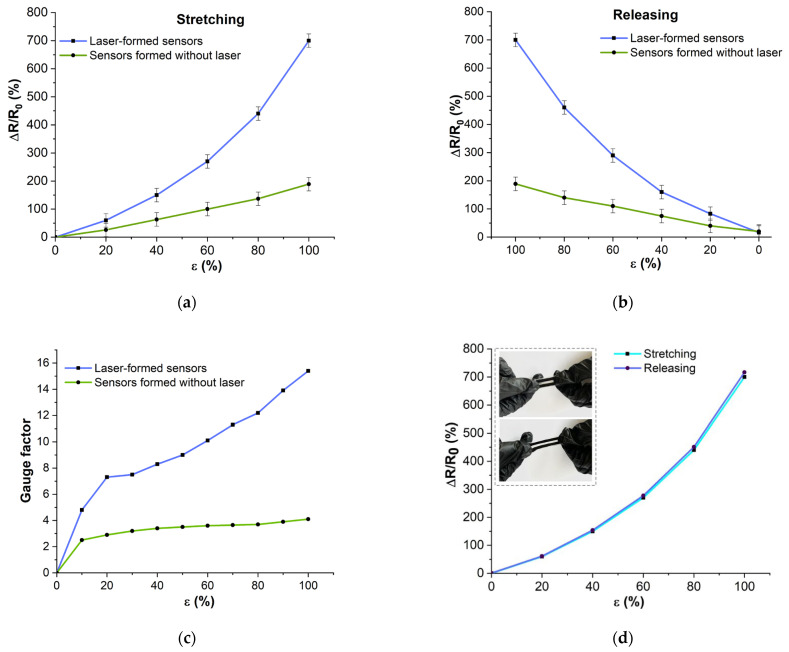
Deformation response of the developed sensors with and without laser exposure: (**a**) dependence of the relative change of resistance during stretching; (**b**) dependence of the relative change of resistance during releasing; (**c**) gauge factor at different strains; (**d**) dependence of the relative change of resistance during tension and release (hysteresis).

**Figure 10 micromachines-14-01106-f010:**
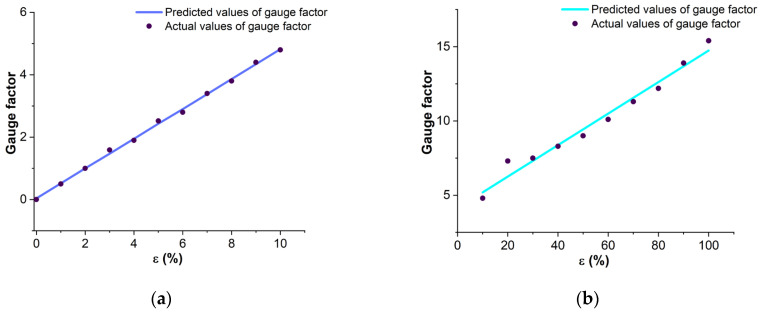
Linearity determination of the GF of the developed sensors where: (**a**) the comparison of predicted GF values with actual values at the area ε < 10%; (**b**) the comparison of predicted GF values with actual values at the area ε > 10%.

**Figure 11 micromachines-14-01106-f011:**
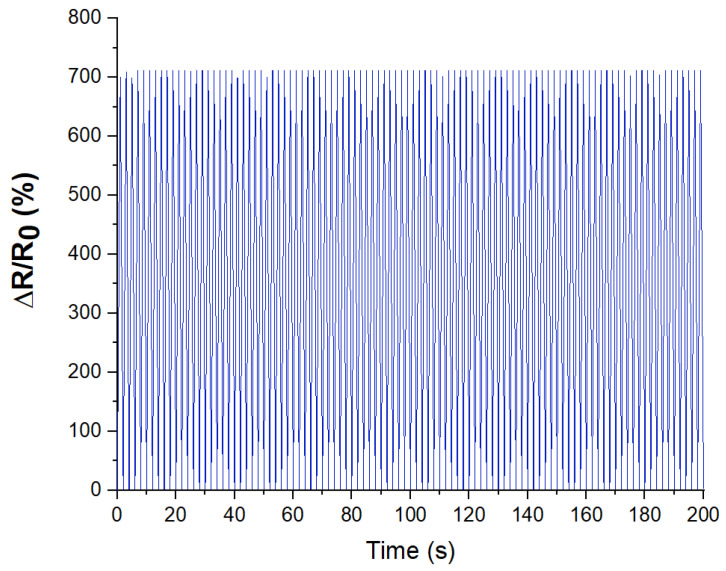
Sensor cyclic load graph: stretching ε = 100% and relaxation to the original length within the first 200 s.

**Figure 12 micromachines-14-01106-f012:**
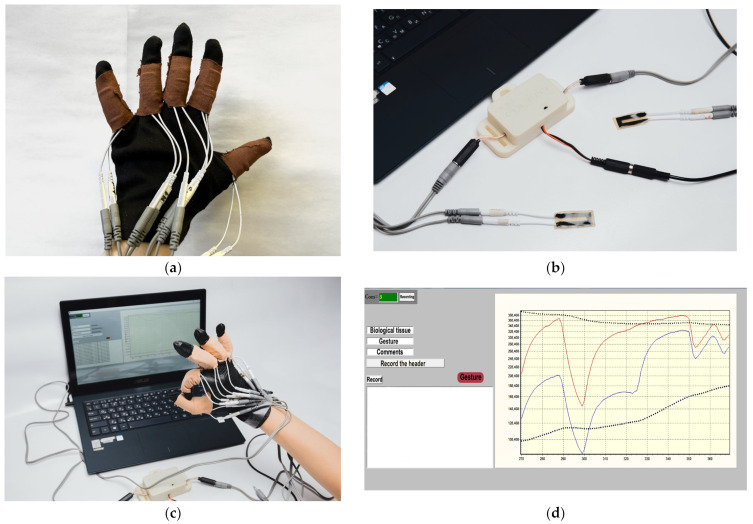
Developed sensors, and a smart system based on them, to recognize hand gestures: (**a**) sensors based on carbon nanotubes and elastomers; (**b**) glove with five installed carbon nanotube and elastomeric sensors for gesture recognition applications; (**c**) a smart gesture recognition system consisting of a glove with sensors and an electronic unit connected to a computer; (**d**) the program for visualization, reading and storing data from the sensors.

**Figure 13 micromachines-14-01106-f013:**
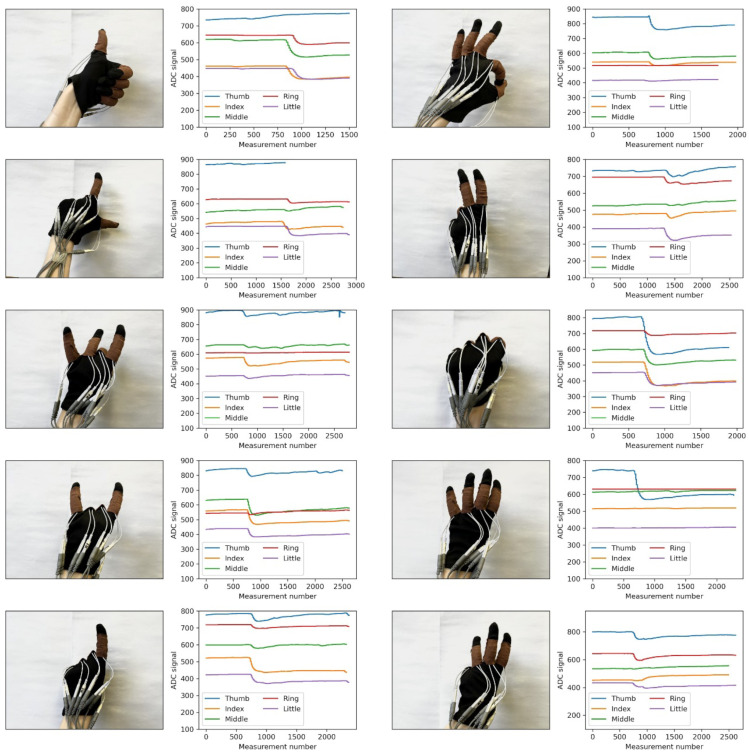
Dependence of resistive sensor readings from five fingers for 10 recognizable gestures.

**Figure 14 micromachines-14-01106-f014:**
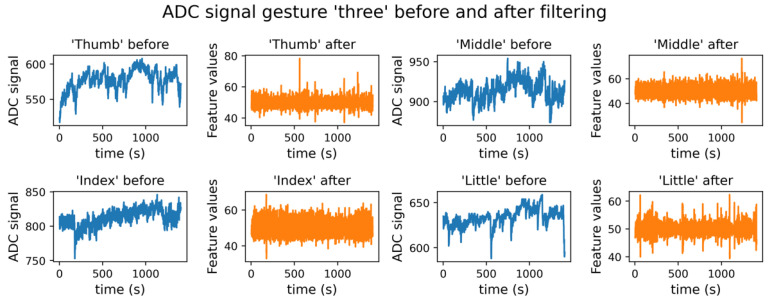
Data in a demonstration gesture “three”, before (f) and after filtering ($\hat{f}$) for index, middle, little fingers, and the thumb.

**Figure 15 micromachines-14-01106-f015:**
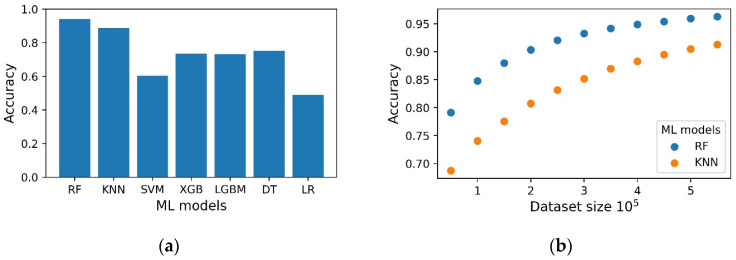
Accuracy for the ML models KNN, SVM, RF, XGB, LGBM, DT (**a**) during training on dataset of size 10^5^ rows. (**b**) Accuracy for ML models RF, KNN during training on a part of the dataset of size 0.5 × 10^5^ to 5.5 × 10^5^.

**Table 1 micromachines-14-01106-t001:** Parameters of laser exposure.

Parameter	Value
Wavelength	1064 nm
Energy density	0.5 J/cm^2^
Pulse duration	100 ns
Frequency	30 kHz
Beam speed	450 mm/s

**Table 2 micromachines-14-01106-t002:** Comparison of the characteristics of sensors.

Method	Initial Resistance, kOhm	Young’s Modulus, kPa	Gauge Factor	Strength, kPa	Hysteresis, %
Sensors formed without laser	19	40	2.5–4.1	827	5
Laser-formed sensors	3	47	4.8–15.4	963	3

## Data Availability

The data taken with the developed smart system, which was used to train the gesture recognition can be found on the following link: https://github.com/DmitriiRybkin/Gesture_recognition (accessed on 28 April 2023).

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
