# Peer review of "Laser-Formed Sensors with Electrically Conductive MWCNT Networks for Gesture Recognition Applications"

_micromachines, 2023, doi:10.3390/mi14061106_

Round 1
Reviewer 1 Report
In this work, the authors propose a method for forming an electrically conductive network with a microsystem for gesture recognition applications. From the sensor fabrication to human trial to the ML analysis, this decent work with many detailed experiments looks good for this journal. Authors need to address the following questions before acceptance.
1. Authors need to explain why they made the U shape design of this sensor. What is the thickness of the sensor? How to determine the ratio of the CNT and PDMS for the best sensing performance?
2. How to improve the signal-to-noise ratio of the sensor in Figure 14?
3. In Figure 13, the x and y label are missing. Authors need to explain more about this experiment. In addition, it is better to show several quick gestures in one figure to prove the repeatability and responsibility.
Author Response
Response to Reviewer 1 Comments
In this work, the authors propose a method for forming an electrically conductive network with a microsystem for gesture recognition applications. From the sensor fabrication to human trial to the ML analysis, this decent work with many detailed experiments looks good for this journal. Authors need to address the following questions before acceptance.
The authors thank you for your appreciation of the work and accurate comments.
Point 1: Authors need to explain why they made the U shape design of this sensor. What is the thickness of the sensor? How to determine the ratio of the CNT and PDMS for the best sensing performance?
Response 1: Thank you for correct observations. U-shape form of sensors was chosen for the convenience of the wires, which were added to the MWCNT/PDMS composite before the PDMS solidified. In addition, the U-shaped design has a longer active material length than the I-shaped sensor. This provides a greater change in resistance when deformed. The wires were placed on one end of the sensor due to the chosen shape. The thickness of the formed active material layer was 0.5 mm. This thickness is mainly due to the magnitude of the laser radiation drag to form an electrically conductive nanostructured layer. To increase the tensile strength and insulation of the conductive layer, the sensor was coated with 0.25 mm thick layers of PDMS on top and bottom. The total thickness of the sensor was therefore 1 mm.
Text was added to Lines 188-197.
The ratio of the CNT and PDMS for the best sensing performance was determined experimentally in section 3.1. SWCNT concentrations of 2%, 3%, 4% were investigated. Increasing the MWCNT concentration above 4 wt% significantly increases the viscosity of the MWCNT/PDMS blend. This makes it difficult to homogenize the MWCNT in PDMS during the fabrication of the active material. Concentrations of CNT below 2 wt% will cause the sensor resistance to rise above 1 MOhm. Such high resistance values result in difficulty in detecting changes in resistance and are unacceptable for sensor construction. Therefore, the effect of MWCNT concentration on the resistance of the active sensor material was investigated within a MWCNT concentration range of 2-4 wt%.
Text was added to Lines 387-391.
Point 2: How to improve the signal-to-noise ratio of the sensor in Figure 14?
Response 2: We appreciate your comment. Figure 14 shows the application of filtering to the sensor signal. As the sensor is used over a period of time, a change in the baseline resistance of the sensor occurs. Then there is practically no change in baseline resistance. Filtering is used to correct for the change in the sensor resistance baseline (blue curves before filtering, orange after). Signal fluctuations in the orange curves are the result of a change in sensor resistance and are not noise.
We have added more information to the text (Lines 526-531).
Point 3: In Figure 13, the x and y label are missing. Authors need to explain more about this experiment. In addition, it is better to show several quick gestures in one figure to prove the repeatability and responsibility.
Response 3: Thank you for paying attention to the details. We have corrected Figure 13. The signal capture for the long-term measurement is shown in Figure 14.
We expanded the text of the article with a more detailed description of the gesture recognition experiment (Lines 507--513). The subject wore a glove with sensors. The wires that come from the sensors were connected to the electronic unit to record the signals. The electronic unit was connected to a computer and ran the developed software, which recorded the data received from the sensors over time in a separate file. Eleven different gestures were chosen for the gesture recognition experiment, including a basic gesture. Such gestures were medically necessitated by the fact that rehabilitation using various gestures is often used in restoring communication in aphasia [https://doi.org/10.1080/02687038.2013.805726]. The signals produced by the alternation between the base gesture and the studied gesture were recorded. The basic gesture was a hand clenched in a fist.
Changes in manuscript are highlighted in orange.

Reviewer 2 Report
In this manuscript, authors propose a method for manufacturing the flexible strain sensor based in conductive MWCNT modified by laser irradiation, and demonstrate that the sensor has the potential for gesture recognition. The analysis made in the manuscript is clear for understanding the subject. This work is recommended to be published before the following observations are addressed.
1) The novelty is not very clear. The authors should highlight this point.
2) The framework of this manuscript is not very reasonable. In section 2, it takes a great amount of space to describe the characterization of the sensor and the establishment of the gesture recognition system, while the related data is mentioned in section 3.
3) The methods of manufacturing are not very clear. What’s the substrate used when doing screen-printing? And after laser irradiation, how is the printed pattern transferred to PDMS encapsulation layer?
4) Fig.2 (a) is not so good. The pattern and the parameters should be presented separately, and latter could be presented in a table. In addition, it is not suggested to take a photo of the screen.
5) The description of Fig.2 (b) is not clear. In the picture, something green, red, and white could be seen, but there are no notes for them. Besides, the background of picture is a little messy, which could not highlight the sensor itself.
6) In line 179 of page 5, “figure 2” should be adjusted as “figure 3”. Similar problems exist in line 489 of page 16, and line 511 of page 17.
7) In section 2.2, the last paragraph could be combined with the first paragraph, which has the similar content.
8) In Fig.7, the enlarged region should be marked in the picture with smaller scale.
9) The last paragraph in page 10, which introduces the effect of laser, should be placed in section 1.
10) The information presents in Table 1 can be also obtained in Fig.8.
11) In Fig.12 (d), the content displays by the PC is not very clear.
12) Laser processing has been applied to fabricate various flexible sensors. The authors may refer to recent publications. 1. Opto-Electronic Advances, 2023: 220172-1-220172-10; 2. Nano Energy, 2022, 103: 107803; 3. ACS Applied Materials & Interfaces, 2022, 14(38): 43877-43885.
NA
Author Response
Response to Reviewer 2 Comments
In this manuscript, authors propose a method for manufacturing the flexible strain sensor based in conductive MWCNT modified by laser irradiation, and demonstrate that the sensor has the potential for gesture recognition. The analysis made in the manuscript is clear for understanding the subject. This work is recommended to be published before the following observations are addressed.
We are very grateful for your attention to our work.
Point 1: The novelty is not very clear. The authors should highlight this point.
Response 1: Thank you for the valuable note! We highlight the novelty of the proposed solutions in the text (Lines 145-149)
The novelty of the project lies in the use of laser exposure to form electrically conductive networks of carbon nanotubes for the manufacture of gesture-registering strain sensors. This approach significantly increases the conductivity and sensitivity of the strain sensors, allowing the use of lower MWCNT concentrations. Thus, the formed percolation network of MWNTs is more electrically sensitive to mechanical deformations.
Point 2: The framework of this manuscript is not very reasonable. In section 2, it takes a great amount of space to describe the characterization of the sensor and the establishment of the gesture recognition system, while the related data is mentioned in section 3.
Response 2: Thank you for your observation! Indeed, we have tried to follow the template of the journal structure by listing in the second section the materials used, manufacturing methods and test methods, and in the third section the accompanying data and results.
Point 3: The methods of manufacturing are not very clear. What’s the substrate used when doing screen-printing? And after laser irradiation, how is the printed pattern transferred to PDMS encapsulation layer?
Response 3: The methods of sensor manufacturing were as follows. A layer of MWCNT/PDMS was applied to the substrate through a U-shape screen. The forms printed by a 3d printer were used as both the substrate and the screen. After the active layer was cured, it was exposed to laser treatment at a selected parameters. After the MWCNT/PDMS layer was screen-printed and laser processed, a thin layer of PDMS was poured on top through another rectangular screen. The active material layer with the insulating PDMS layer was then inverted and the PDMS layer was poured on the other side in the same shape.
Thus, the methods of sensors manufacturing have been clarified in the text (Lines 186-188, Lines 198-206).
Point 4: Fig.2 (a) is not so good. The pattern and the parameters should be presented separately, and latter could be presented in a table. In addition, it is not suggested to take a photo of the screen.
Response 4: Thank you for the correct point. Fig.2 (a) has been replaced by a better one. The pattern and the parameters of laser treatment were separated. The parameters were transferred to Table 1. Corresponding changes have been made to the text in section 2.2.
Point 5: The description of Fig.2 (b) is not clear. In the picture, something green, red, and white could be seen, but there are no notes for them. Besides, the background of picture is a little messy, which could not highlight the sensor itself.
Response 5: Thank you for your expert assessment and for your attention to detail. The authors apologise for the poor quality of the figure. Fig.2 (b) has been replaced by a better one. A description of the figure has been added.
Point 6: In line 179 of page 5, “figure 2” should be adjusted as “figure 3”. Similar problems exist in line 489 of page 16, and line 511 of page 17.
Response 6: Thank you very much for your consideration. Checked and corrected the picture references throughout the article
Point 7: In section 2.2, the last paragraph could be combined with the first paragraph, which has the similar content.
Response 7: Thank you for the recommendation, we have merged the above paragraphs (Lines: 175-176)
Point 8: In Fig.7, the enlarged region should be marked in the picture with smaller scale.
Response 8: We are grateful for your comment, which has improved the quality of the work. Fig.7c was taken from an area not shown in Fig.7b. Therefore, Fig.7c has been replaced by a new one. In Fig.7b the enlarged areas (Fig.7c and d) were highlighted. The description for Fig.7 has been changed.
Point 9: The last paragraph in page 10, which introduces the effect of laser, should be placed in section 1.
Response 9: Thank you very much, we have listened to your comment and relocated the paragraph (Lines: 122-141).
Point 10: The information presents in Table 1 can be also obtained in Fig.8.
Response 10: Thank you for your comment! We have removed Table 1, keeping only Figure 8, which clearly shows the results.
Point 11: In Fig.12 (d), the content displays by the PC is not very clear.
Response 11: Thank you for your comment! Figure 12c shows a general view of the developed system. Due to the fact that the content displayed by the computer in the picture cannot be made clearer, we included a screenshot of the screen and showed it in Figure 12d. We changed the arrangement of the pictures and added more clarification (Lines 495-498). The visualisation software gave a graph of the amplitude of the ADC values over time. A screenshot of the software window is shown in Figure 12d.
Point 12: Laser processing has been applied to fabricate various flexible sensors. The authors may refer to recent publications. 1. Opto-Electronic Advances, 2023: 220172-1-220172-10; 2. Nano Energy, 2022, 103: 107803; 3. ACS Applied Materials & Interfaces, 2022, 14(38): 43877-43885.
Response 12: Thank you for recommending that we consider very interesting articles. We are pleased to include them in our review: Line 145.
Changes in manuscript are highlighted in green.

Reviewer 3 Report
This work reported a laser-assisted method to form conductive MWCNT network in PDMS elastomer to fabricate stretchable sensors for gesture recognition applications. The process of laser scribing was proven to significantly enhance the electrical conductivity and sensitivity of the strain sensor, which can promote the use of less amount of MWCNT. The author also demonstrated the use of MWCNT-based strain sensors in wearable gesture recognition. This work provided a very detailed investigation of the fabrication, properties, and applications of the newly formed sensors. Minor concerns should be addressed before the publication.
1. Was MWCNT/PDMS screen-printed or stencil-printed? Figure 1 shows stencil printing but the description in the manuscript shows screen printing. These are two different methods. Please clarify.
2. What is the dimension of the sensor electrode? It seems the electrode was very thick. Scale bar should be provided.
3. The effect of MWCNT with different concentrations on the resistance of the sensor was studied. The reviewer is also wondering if the power of the laser could affect the resistance. If 2% MWCNT was treated with a higher-power laser, will lower resistance be achieved.
4. The transmission of gesture signals is wireless, but the sensors include lots of wire. What are their functions?
5. The final sensor prototypes look very bulky, although they show promising sensibility and applications. Is there any method to reduce the dimension of the sensor?
Author Response
Response to Reviewer 3 Comments
This work reported a laser-assisted method to form conductive MWCNT network in PDMS elastomer to fabricate stretchable sensors for gesture recognition applications. The process of laser scribing was proven to significantly enhance the electrical conductivity and sensitivity of the strain sensor, which can promote the use of less amount of MWCNT. The author also demonstrated the use of MWCNT-based strain sensors in wearable gesture recognition. This work provided a very detailed investigation of the fabrication, properties, and applications of the newly formed sensors. Minor concerns should be addressed before the publication.
We appreciate your opinion and constructive criticism.
Point 1: Was MWCNT/PDMS screen-printed or stencil-printed? Figure 1 shows stencil printing but the description in the manuscript shows screen printing. These are two different methods. Please clarify.
Response 1: Thanks for the comment! We apologise for this misprint. We have corrected figure 1 according to the text of the article.
Point 2: What is the dimension of the sensor electrode? It seems the electrode was very thick. Scale bar should be provided.
Response 2: Thank you for the feedback. We have added photos of the sensors with a scale bar for better visibility in Figure 6 (b,c,d). In addition, we have supplemented section 2.2 with a description of the screen template used (Lines 188-189). The thickness of the sensor electrode is limited by it, but can be modified in the future to suit the required applications.
Point 3: The effect of MWCNT with different concentrations on the resistance of the sensor was studied. The reviewer is also wondering if the power of the laser could affect the resistance. If 2% MWCNT was treated with a higher-power laser, will lower resistance be achieved.
Response 3: We have investigated this issue in our experiments. If 2% MWCNT is treated with a higher power laser, the silicone that is in the active layer of the sensor overheats and burns out. As a result this degrades the performance of the sensor. It was also shown in earlier works of the authors [https://doi.org/10.3390/polym14091866; https://doi.org/10.1016/j.compstruct.2020.113517 et al.] that when forming a percolation network of nanotubes on a silicon substrate or in a polymer matrix it is necessary to determine a threshold value of energy density. Such a value will contribute to the compromise of the maximum number of percolation nodes and the minimum sublimation of defective nanotube regions under the action of laser radiation.
We have added these results to the manuscript text (Lines 388-390). Thank you for your interest in our work!
Point 4: The transmission of gesture signals is wireless, but the sensors include lots of wire. What are their functions?
Response 4: Thank you for your thoughtful critique. It is true that gesture signals are transmitted wirelessly, as the electronic unit is equipped with a bluetooth module. The sensors contain wires for connection to the electronic unit. The function of the wires is to transmit the signal from the active sensor layer to the electronic unit. The electronic block can be powered by a USB cable, or it can be powered by a battery pack. Then the unit can be attached to the belt/person and data can be collected remotely, in any position of the person. We have explained this point in the text (Lines 499-501).
Point 5: The final sensor prototypes look very bulky, although they show promising sensibility and applications. Is there any method to reduce the dimension of the sensor?
Response 5: This is a very important observation. The size of the sensor can be adjusted by selecting a new template. Nevertheless, the chosen dimensions for the application described in the manuscript are appropriate because they capture joint movements. We have supplemented the text with this note (Lines 188-189).
Changes in the manuscript are highlighted in blue.
